# Filamentous structures in the cell envelope are associated with bacteroidetes gliding machinery

Satoshi Shibata [1,7 ✉], Yuhei O. Tahara[2,3], Eisaku Katayama[2,8], Akihiro Kawamoto[4,9], Takayuki Kato [4,9], Yongtao Zhu[5,10], Daisuke Nakane [6], Keiichi Namba [4], Makoto Miyata[2,3], Mark J. McBride [5] & Koji Nakayama [1 ✉]

Many bacteria belonging to the phylum Bacteroidetes move on solid surfaces, called gliding motility. In our previous study with the Bacteroidetes gliding bacterium *Flavobacterium johnsoniae*, we proposed a helical loop track model, where adhesive SprB filaments are propelled along a helical loop on the cell surface. In this study, we observed the gliding cell rotating counterclockwise about its axis when viewed from the rear to the advancing direction of the cell and revealed that one labeled SprB focus sometimes overtook and passed another SprB focus that was moving in the same direction. Several electron microscopic analyses revealed the presence of a possible multi-rail structure underneath the outer membrane, which was associated with SprB filaments and contained GldJ protein. These results provide insights into the mechanism of Bacteroidetes gliding motility, in which the SprB filaments are propelled along tracks that may form a multi-rail system underneath the outer membrane. The insights may give clues as to how the SprB filaments get their driving force.

[1] Department of Microbiology and Oral Infection, Graduate School of Biomedical Sciences, Nagasaki University, Nagasaki, Japan. [2] The OCU Advanced Research Institute for Natural Science and Technology (OCARINA), Osaka Metropolitan University, Sumiyoshi-ku, Osaka, Japan. [3] Graduate School of Science, Osaka Metropolitan University, Sumiyoshi-ku, Osaka, Japan. [4] Graduate School of Frontier Biosciences, Osaka University, Suita, Osaka, Japan. [5] Department of Biological Sciences, University of Wisconsin-Milwaukee, Milwaukee, WI 53201, USA. [6] Department of Engineering Science, Graduate School of Informatics and Engineering, The University of Electro-Communications, Chofu, Tokyo, Japan. [7] Present address: Division of Bacteriology, Department of Microbiology and Immunology, Faculty of Medicine, Tottori University, Yonago, Tottori, Japan. [8] Present address: Waseda Research Institute for Science and Engineering, Okubo Shinjyuku, Tokyo, Japan. [9] Present address: Institute for Protein Research, Osaka University, Suita, Osaka, Japan. [10] Present address: Department of Biological Sciences, Xi'an Jiaotong-Liverpool University, Suzhou, Jiangsu, China. ✉email: sshibata@tottori-u.ac.jp; knak@nagasaki-u.ac.jp

Cells of many bacterial species belonging to the phylum Bacteroidetes move over surfaces at ~1–5 μm per second in a process called gliding motility[1]. Gliding motility requires cell contact with a solid surface, since cells suspended in liquid do not actively move. Genetic analyses have suggested that the Bacteroidetes gliding motility is unrelated to other well-studied bacterial motility mechanisms such as flagellar motility, type IV pilus-mediated twitching motility, myxobacterial gliding motility, and *Mycoplasma* gliding motility, but instead relies on novel machinery consisting of Gld and Spr proteins that are confined to members of the large and diverse phylum Bacteroidetes such as *Flavobacterium johnsoniae*[2–6]. Some Gld and Spr proteins are not only involved in gliding motility but are also components of the type IX secretion system (T9SS)[7]. T9SSs were first identified and studied in the nonmotile oral pathogen *Porphyromonas gingivalis* and in *F. johnsoniae*[8–11]. Recent studies have revealed that T9SS is a unique system that is clearly different from other secretion systems in terms of its supramolecular structure and secretion mechanism[12–14]. *F. johnsoniae* cells use the T9SS to secrete dozens of proteins, including soluble extracellular enzymes and motility adhesins that reside on the cell surface[15–17].

*F. johnsoniae* has been studied as a model organism to understand the molecular mechanism of Bacteroidetes gliding motility. Genetic and molecular analyses demonstrated that SprB, a huge filamentous 6497 amino acid protein, is a primary cell-surface adhesin of *F. johnsoniae*[18]. Immunofluorescence microscopic analysis using antiserum against SprB revealed that the SprB filaments are propelled at ~2 μm per second with a ~19° tilt with respect to the long axis of the cell. Mathematic analyses suggested a helical-loop track model in which gliding motors act on SprB filaments that have attached to a surface-generating rotation and translocation of the cell body[19]. A recent cryo-EM study has revealed that GldL and GldM, a transmembrane core complex for gliding motility has a structural organization similar to that of the bacterial flagellar stator complex consisting of MotA and MotB, which acts as a proton channel for torque generation[20–22]. GldLM complex shows relatively static localization distributed in multiple foci along the cell body and fueled by the proton gradient to drive the helical motion of SprB adhesin[23,24]. However, the structural basis for why SprB moves along a helical track is still unknown.

In this paper, a detailed analysis of SprB movement on the *F. johnsoniae* cell surface by immunofluorescence microscopy and morphological analysis of the gliding machinery by electron microscopy suggests the possibility of a multirail structure as a component of the helical-loop track involved in SprB movement. Similar multirail structures were seen in the distantly related gliding Bacteroidetes *Saprospira grandis*, suggesting that this may be a general feature of the Bacteroidetes gliding machinery. The results provide insights into the mechanism of this common form of bacterial gliding motility.

## Results

**Counterclockwise rotation of a gliding cell**. In our previous study, we proposed a helical-loop track model for gliding motility of *F. johnsoniae* where SprB adhering to a substratum was propelled along a helical-loop track on the cell surface, resulting in rotation and translocation of the cell[19]. Here, we investigated whether gliding cells actually rotate as proposed. To visualize cell rotation, we attempted to find a cell-surface protein that did not change position with respect to the long axis during gliding, and observe its behavior by total internal reflection fluorescence (TIRF) microscopy using fluorescently labeled antiserum. Cell-surface proteins were isolated and collected by centrifugation. The

isolated cell-surface proteins were separated by SDS-PAGE (Supplementary Fig. 1a). SprB with molecular mass greater than 250 kDa was detected by immunoblot analysis using anti-SprB antiserum (Supplementary Fig. 1b). Major protein bands on the gel were identified by peptide mass fingerprinting analysis using MALDI-TOF-MS, which revealed that the cell-surface protein fraction contained proteins with putative outer membrane protein (OMP) domains (Fjoh_0697, Fjoh_1311, and Fjoh_3514) and putative TonB-dependent receptors (Fjoh_0403, Fjoh_0736, Fjoh_4221, and Fjoh_4559) (Supplementary Fig. 1a and Supplementary Table 1). Amino acid sequence analyses for predicting signal peptides by SignalP 4.0 software[25] and for predicting protein localization by CELLO v2.5 subcellular localization predictor[26,27] suggested that most of the proteins were located at the outer membrane (Supplementary Table 1). We generated rabbit antiserum against the cell-surface protein fraction. Immunofluorescence microscopy using this antiserum and Alexa Fluor 555-conjugated secondary antibodies showed that signals were dispersed on the cell surface (Supplementary Fig. 1c). TIRF microscopy with a translocating cell revealed that there were two types of signal movement (Supplementary Movie 1). Type I signals, the most common, were located at a position a fixed distance away from a cell pole and periodically appeared on the surface with respect to the short axis when the cell moved on a substratum, indicating cell rotation. TIRF analysis showed that the type I signals always appeared from one side with respect to the short axis of a cell moving in one direction, demonstrating rotation in a counterclockwise (CCW) direction when viewed from the rear to the advancing direction of the cell (Supplementary Movie 2). Type II signals, which comprised less than 20% of total signals, moved on the cell surface with respect to both axes like the SprB signal (Supplementary Movie 3). This type of signal behavior was expected since SprB, which migrates from one pole to the other along an apparent helical track[19], was present in the surface protein fraction that was used to generate the polyclonal antiserum. We also generated antiserum against purified OMP domain-containing protein Fjoh_0697, which appeared to be a major cell-surface protein (Supplementary Fig. 1a). TIRF analysis using anti-Fjoh_0697 antiserum revealed that the Fjoh_0697 signal did not migrate from pole to pole but was instead located at a position a fixed distance from a cell pole. In a translocating cell Fjoh_0697 periodically appeared on the surface with respect to the short axis, demonstrating CCW rotation of the gliding cell as observed from the rear of the cell (Fig. 1a and Supplementary Movie 4).

SprB is the primary motility adhesin on the *F. johnsoniae* cell surface. The left-handed helical flow of immunolabeled SprB on a gliding cell was previously observed[19]. In this study, for detailed analysis of the individual SprB signal movement on a gliding cell surface, limited numbers of SprB molecules on a cell were labeled with highly diluted antiserum (see "Methods") to reduce signals and track SprB signals easily, and movement of signals on the surface of gliding cells was observed by fluorescence microscopy. Under these conditions, about 58% of cells had 1–10 SprB signals and the rest had none. Consistent with the previous report[19], the SprB signals were propelled between cell poles along an apparently helical-loop track on gliding *F. johnsoniae* cells (Fig. 1b and Supplementary Movie 5). Cells migrated 6.1 ± 1.1 μm (mean ± standard deviation, N = 63) when the Fjoh_0697 signal made one revolution in the direction of the short axis of a cell (helical pitch of Fjoh_0697) (Fig. 1c left, d), while the moving distance of SprB on the substratum during one revolution of SprB in the direction of the short axis of a cell (apparent helical pitch of SprB) (Fig. 1c right, d) was 4.55 ± 1.1 μm (N = 22). Note that the helical pitch was measured by the SprB signals moving from the lagging pole to the leading pole. These signals followed CCW

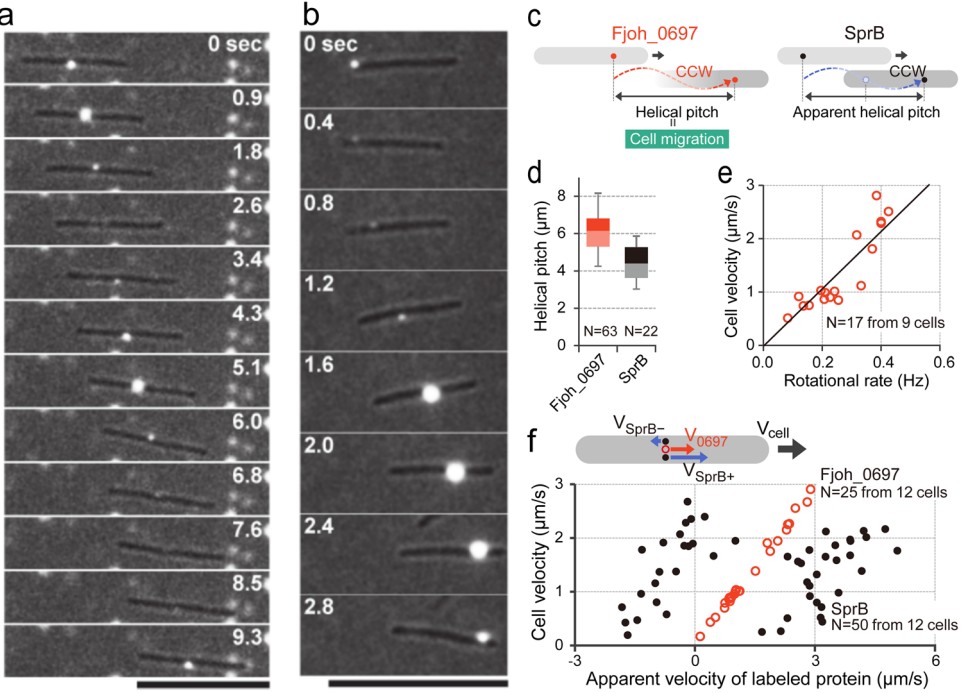

**Fig. 1 Rotation of gliding cells.** Gliding cells were labeled with anti-Fjoh_0697 (**a**) or anti-SprB (**b**) antisera, and Alexa Fluor 555-conjugated anti-rabbit IgG secondary antibody. Signals were observed by TIRF microscopy. Cell outlines were visualized by simultaneous weak illumination using a halogen lamp. Bars indicate 10 μm. **c** Schematic illustration of the helical movement of Fjoh_0697 and SprB signals. **d** Length of helical pitch of the movement of Fjoh_0697 and the distance of SprB movement during one revolution on the cell. Box plot represents the minimum value, 25th percentile, median, 75th percentile, and maximum value of the helical pitch. **e** Relation of cell velocity and rotational rate. The rotation rate of a cell was calculated from the movement of Fjoh_0697 signal with respect to the short axis of cell. The fitting is shown with the black line. **f** Relation of the cell velocity and the apparent velocity of labeled proteins Fjoh_0697 (empty circle) and SprB (filled circle). A schematic illustration of the protein movement with respect to the long axis of the cell is shown in the same panel.

trajectories on the view from the rear end. Considering the left-handed helical-loop track, it is expected that the SprB signals moving from the leading pole to the lagging pole follow CW trajectories. However, we could not follow the CW trajectories because these signals moved slowly and the cells switched back their moving direction frequently. During cell migration, the rotational rate of Fjoh_0697 signals was directly proportional to the cell velocity (Fig. 1e). To show the relationship between movements of a cell and surface proteins, apparent velocities of labeled proteins were determined; SprB (50 signals) and Fjoh_0697 (25 signals) from 12 cells were plotted at each cell velocity on a graph (Fig. 1f). Plots showed that the velocity of Fjoh_0697 signals was in close agreement with the cell velocity measured by the positional displacement of the cell shape. These data demonstrated that Fjoh_0697 signals are located at fixed positions on the cells. The SprB signals moving toward the leading pole moved faster than cell velocity (Fig. 1f, VSprB+), whereas the SprB signals moving toward the lagging pole moved slower (Fig. 1f, VSprB−). These data suggest that the former is not attached to the substratum but the latter is dragged due to the interaction with the substratum to propel the cell. The velocity of SprB toward the leading pole and lagging pole was positively correlated in both cases with the cell velocity. However, the correlation was weaker than that of Fjoh_0697. The difference may be accounted for by the fact that the velocity of SprB varies from molecule to molecule, and SprB that advances the cell is only a part of SprB molecules. The movement of SprB on a cell was then examined in more detail.

**Multiple lanes in helical-loop track for SprB movement.** Some SprB signals exhibited unexpected movements such as mid-cell

U-turns. Figure 2a and Supplementary Movie 6 show an SprB signal that moved toward the anterior (right) pole, looped around the anterior pole at ~1.8 s, and then moved toward the posterior pole. Interestingly, before the signal reached the posterior pole, it made a U-turn and moved toward the anterior pole (3.5–4.4 s). The SprB signal then looped around the anterior pole and migrated all of the way to the posterior pole without conducting another mid-cell U-turn (5.2–9.5 s). The frequency of mid-cell U-turns was about 0.07 per min per SprB, suggesting that the U-turn events do not happen frequently. Multiple SprB signals on a cell did not always move at the same speeds. In Fig. 2b and Supplementary Movie 7, a cell that had two SprB signals traveling in the same direction is shown. The low-intensity signal moved more slowly (~1.2 μm per second) than did the high-intensity signal (about 2.4 μm per second). The high-intensity signal thus overtook and passed the low-intensity signal while traveling to the anterior (right) pole. These results suggest the possibility that the track on which SprB traveled had multiple lanes.

**Alteration of SprB velocity during translocation.** Since we observed that some SprB signals moved with different speeds on the cell surface, we asked if SprB can change velocity during pole-to-pole movement. Since it was difficult to track individual SprB signals on cells of normal size for long periods of time, we observed SprB movement on filamentous cells generated by inhibiting cell septation with the antibiotic cephalexin. Addition of 20 μg per ml cephalexin to growth media resulted in filamentous cells that were about 100 μm long. The localization of SprB on glutaraldehyde-fixed cells as observed by immunofluorescence microscopy showed similar distributions on filamentous (cephalexin-treated) cells and on non-elongated control

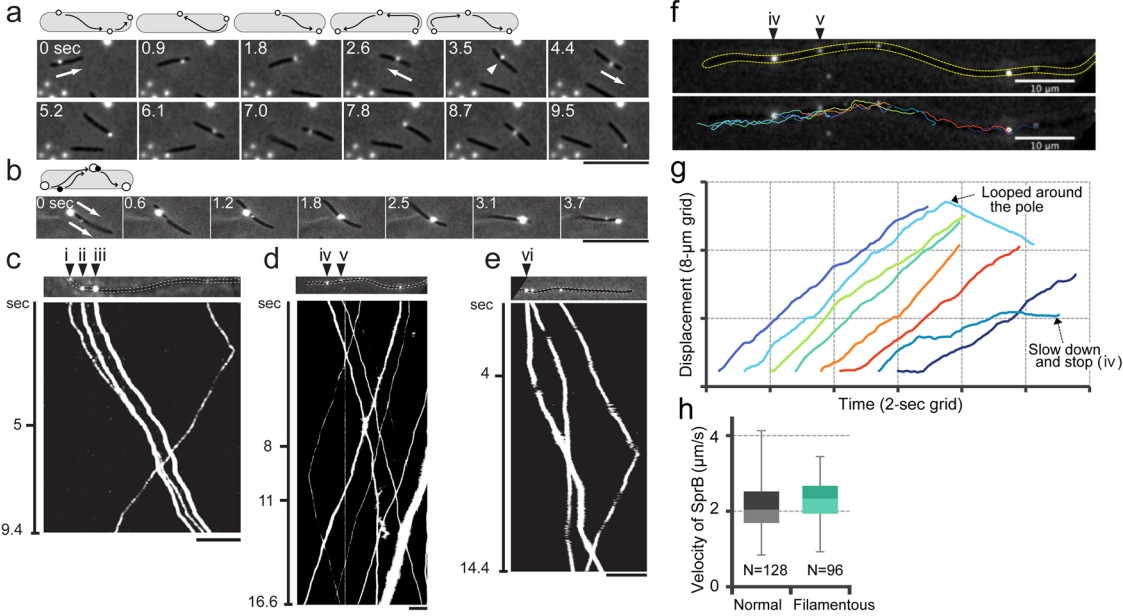

**Fig. 2 Patterns of SprB movement on gliding cells.** Limited numbers of cell-surface SprB proteins were immunolabeled using low-concentration antiserum against SprB and Alexa Fluor 555-conjugated anti-rabbit IgG secondary antibody and observed by phase-contrast fluorescence microscopy. Cell outlines were visualized by simultaneous weak illumination using a halogen lamp. SprB signal movements are depicted by time-lapse montage images and kymographs. **a** U-turn of SprB. **b** Passing of one SprB signal by another. Arrows indicate the direction of movement of SprB signal, and white arrowhead indicates the turn position of SprB signal. **c**–**e** SprB movements on elongated cells. *F. johnsoniae* wild-type cells were grown in the presence of 20 µg per ml of cephalexin for 6 h and labeled with anti-SprB antiserum and Alexa Fluor 555-conjugated anti-rabbit IgG secondary antibody. **c** Kymograph demonstrating passing of one SprB signal by another on a filamentous cell. The kymograph was generated from Supplementary Movie 8, which was taken by phase-contrast fluorescence microscopy. **d** Kymograph of slowdown-and-stop movement of SprB from Supplementary Movie 9. **e** Kymograph of stay-and-go movement of SprB from Supplementary Movie 10. The *x* axis and *y* axis of kymographs represent the positions of SprB signals and elapsed time, respectively. The cell image and the *x* axis of the kymograph in each panel are the same scales. **f** Tracking of SprB movement in panel (**d**). Outline of a filamentous cell is shown by dashed yellow line. Moving trajectories of SprB signals are colored and overplayed on the cell image. **g** Time course of SprB movement along the long axis of cell. Each colored line represents the moving trajectory of SprB with the same color in panel (**f**). Looped around the pole and slowdown-and-stop movements are indicated by arrows. **h** Effect of cephalexin on the velocity of SprB. Box plot represents the minimum value, 25th percentile, median, 75th percentile, and maximum value of the SprB velocity. Bars indicate 10 µm.

cells that had not been exposed to the antibiotic (Supplementary Fig. 2). In both cases, cells had about 1.5 SprB signals per µm on the cell surface under the labeling condition noted in the legend to Supplementary Fig. 2. The average velocity of SprB on a filamentous cell was $2.3 \pm 0.6$ µm per second ($N = 96$), which was similar to SprB movement on a normal-length cell ($2.1 \pm 0.6$ µm per second, $N = 128$) (Fig. 2h). We then determined whether the velocity of SprB along the cell surface varied. SprB movements on filamentous cells that were not themselves translocating on a glass surface are depicted by the kymographs in Fig. 2. On these nontranslocating cells, the SprB molecules were apparently not attached to the substratum, since otherwise the action of the motility motors against these would have resulted in cell movement. The movement of four SprB signals over 9.4 s on a filamentous cell is shown in Fig. 2c (the kymograph from Supplementary Movie 8). Distance between signals marked with ii and iii did not change during the observation, resulting in parallel lines in the kymograph, indicating that these signals moved with similar velocity. In contrast, signal i approached and overtook signal ii within the first 5 s of the recording. Similar to the observation of overtaking movement on a normal cell (Fig. 2b), this result suggests that SprB signals may move with different velocities on a track with multiple lanes. Two other examples of SprB movement on individual filamentous cells are shown in Fig. 2d, e, which correspond to Supplementary Movies 9 and 10, respectively. Parallel oblique lines appeared in these kymographs; however, some lines were not parallel, although SprB signals were moving in the same direction. SprB signals iv and v moved

toward the right with similar velocities of ~3.0 µm per second, resulting in parallel oblique lines for about the first 8 s of observation (Fig. 2d and Supplementary Movie 9). After this point, signal iv moved more slowly and eventually stopped, as indicated by the vertical line in the kymograph after 11 s. As a result, the distance between signals iv and v increased after 8 s. In addition to this slowdown-and-stop behavior, we also observed the stay-and-go movement of SprB signals (Fig. 2e and Supplementary Movie 10). Signal vi did not move until about 4 s after the start of the observation, as indicated by the vertical line. Subsequently, signal vi began to move toward the right and drew an oblique line in the kymograph. In addition, the time course of SprB movement along the long axis of a cell by tracking analysis showed the alteration of SprB velocity during translocation (Fig. 2f, g). These results indicate that the velocity of SprB signals can change during translocation.

**Visualization of multirail structure for gliding machinery.** Observation of SprB movement on the cell surface led us to hypothesize the presence of a multirail structure for the SprB-trains. We visualized such a structure by electron microscopy. Cells were burst by osmotic shock to reduce cell thickness and were then negatively stained with 0.5% uranyl acetate. Possible multirail structures that formed bundles of 2–12 fibers were observed (Fig. 3a–d). The thickness of a fiber in a bundle was $7.5 \pm 0.9$ nm ($N = 58$). The multirail structure was easily peeled from the cell after osmotic shock. Thin filaments appeared to be

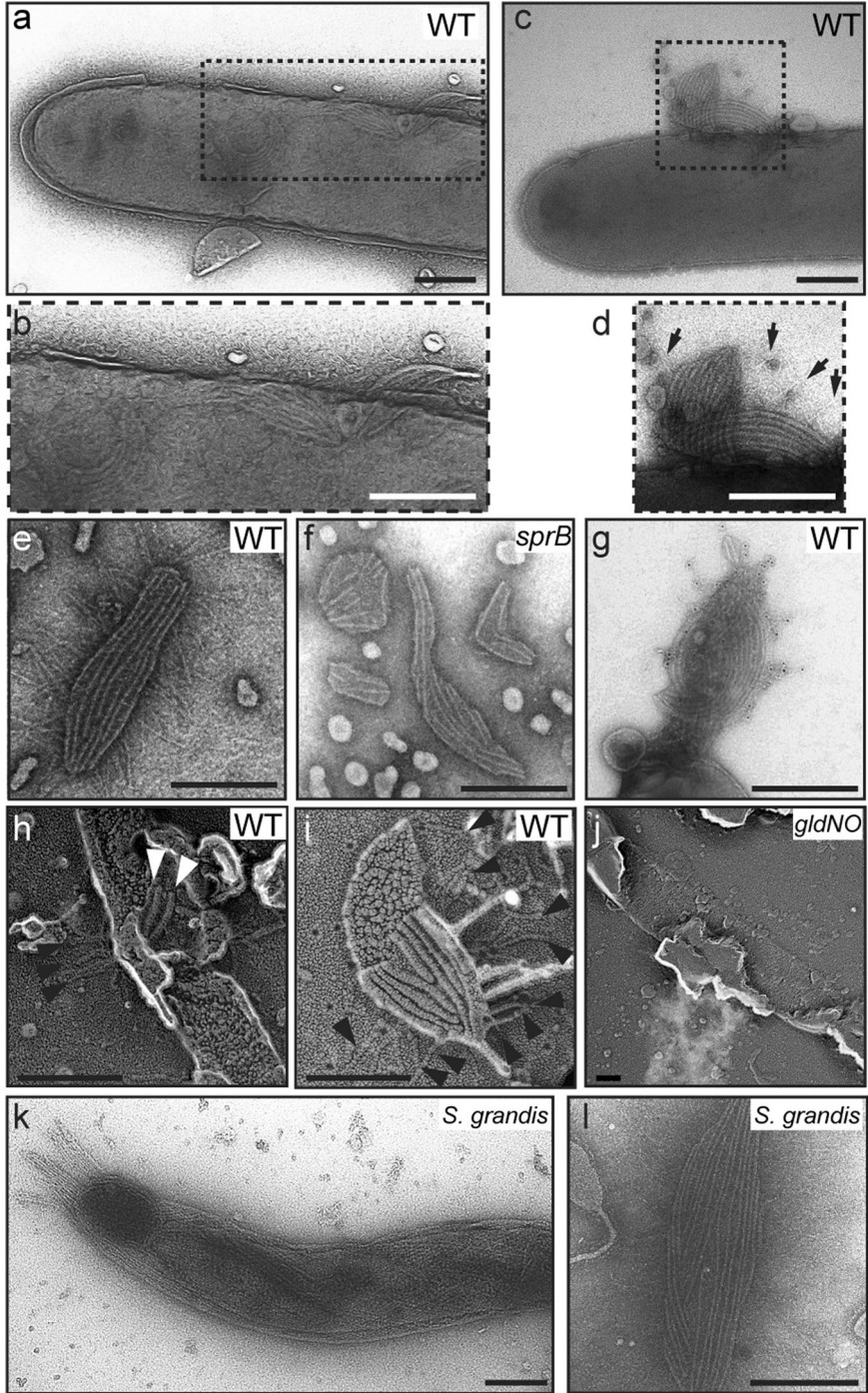

**Fig. 3 Multirail structure underneath the outer membrane. a–d** Osmotically shocked cells of *F. johnsoniae* ATCC 17061 (wild type). Osmotically shocked cells were stained with 0.5% uranyl acetate and observed by TEM. Arrows denote thin filaments associated with the multirail structure. Detached multirail structures collected from the supernatants of osmotically shocked samples of the wild-type (**e**) and *sprB* deletion mutant CJ1922 (**f**) were negatively stained with 2% PTA, pH 7.0. **g** Association of SprB filaments with the multirail structure. Detached multirail structure from the wild-type cells was reacted with anti-SprB antiserum followed by a gold-conjugated secondary antibody and stained with 0.5% uranyl acetate. **h–j** Quick-freeze deep-etch replica images of the periplasmic surfaces of outer membranes of *F. johnsoniae* cells. Cell samples of the wild-type (**h**, **i**) and the *gldNO* deletion mutant CJ1631A (**j**) were deep-etched and rotary-replicated with platinum-carbon. White and black arrowheads indicate the multirail structures attached to the membrane and SprB filaments attached to the flat glass surface, respectively. **k**, **l** Multirail structure in *S. grandis*. Osmotically shocked cells of *S. grandis* were negatively stained with 0.5% uranyl acetate (**k**). Multirail structures, which were peeled off from the cells, were stained with 2% PTA, pH 7.0 (**l**). Bars indicate 200 nm.

attached to the multirail structure (Fig. 3d). Mutants deficient in gliding-related proteins (Gld proteins, Spr proteins, and RemA protein) were examined for the presence of the multirail structure. The multirail structures were observed in each of the *spr* and *remA* mutants at more than 30% of osmotically shocked cells, but

the structures were not observed in any of the *gld* mutants when at least 50 osmotically shocked cells of each *gld* mutant were examined (Supplementary Fig. 3 and Supplementary Table 2). These results suggest that the Gld proteins are required for the formation of the multirail structure. *F. johnsoniae gld* mutants are

completely deficient in gliding[28], whereas *spr* mutants retain limited ability to glide on some surfaces[29,30]. The absence of the rails in the *gld* mutants suggested that these structures may play an important role in gliding.

**Association of the multirail structure with SprB filaments**. For more detailed structural analysis of the multirail structure, the supernatant of an osmotically shocked sample was centrifuged and the resulting precipitate was suspended and analyzed by TEM. Consistent with the observation of multirail structures on osmotically shocked cell surfaces, thin filaments were attached to the multirail structures released from wild-type cells, whereas there were no filaments attached to the multirail structure from the *sprB* mutant (Fig. 3e, f). Immunoelectron microscopic analysis with anti-SprB antiserum revealed that the thin filaments were SprB (Fig. 3g). These results suggest that the multirail structure is associated with the SprB filaments and may form part of the gliding machinery.

**The multirail structure underneath the outer membrane**. The multirail structure was visualized by quick-freeze deep-etch electron microscopy, which provides images of membrane-attached or -embedded structures on a fractured surface. Figure 3h–j shows images of a quick-freeze deep-etch replica with the wild type and *gldNO* mutant treated with osmotic shock. In the wild-type cells, possible multirail structures were observed on the periplasmic surface of the outer membrane and filaments (presumably SprB) extended from the outer membrane (Fig. 3h, i), as previously observed[31]. In contrast, neither multirail structures nor SprB filaments were observed in cells of the *gldNO* mutant (Fig. 3j). To determine the presence of the multirail structure in an intact cell, plunge-frozen *F. johnsoniae* cells were analyzed by cryo-electron tomography. Similar to the previous observation by Liu et al.[32], section images from two representatives of 3D reconstruction of the wild-type cells showed SprB filaments extending from the outer membrane (Fig. 4a, b and Supplementary Movies 11 and 12). In addition, analysis of stacks of section images showed the presence of apparently left-handed, parallel filamentous structures in the periplasm (Fig. 4c, d). However, multirail structures similar to those observed in osmotically burst cells by TEM and quick-freeze deep-etch electron microscopy were not observed by cryo-electron tomography. Neither cell-surface SprB filaments nor the parallel filamentous structures were observed in cells of the *gldJ, gldK, gldL, gldM,* or *gldNO* mutants (Supplementary Movies 13–17). These mutants are completely nonmotile and are also completely deficient in secretion of SprB protein[15,16,33].

**Components of the multirail structure**. Since the mutant study revealed that Gld proteins are required for the formation of the multirail structure, we examined whether Gld proteins were components of the multirail structure by immunoelectron microscopy of osmotically shocked cells using antisera against Gld lipoproteins (GldB, GldD, GldH, and GldJ). Gold particles accumulated on the multirail structure only when anti-GldJ antiserum was used (Fig. 5a), suggesting that GldJ is a component of this structure.

Cells of the *gldA, gldB, gldD, gldF, gldG, gldH, gldI, gldJ, gldK, gldL, gldM,* and *gldNO* mutants showed no multirail structures (Supplementary Fig. 3). GldJ was reported to be unstable in *gldA, gldB, gldD, gldF, gldG, gldH,* and *gldI* mutants[33]. In contrast, GldJ was apparently stable in cells with mutations in *gldK, gldL, gldM,* and *gldN*[28]. We reexamined the amount of GldJ in *gldK, gldL, gldM,* and *gldNO* mutants and obtained similar results, except for a partial reduction of GldJ levels in a *gldK* mutant (Fig. 5b).

We also demonstrated that *sprB* mutant cells retained wild-type levels of GldJ protein (Fig. 5b). Blue native-PAGE analyses suggested that GldJ was part of a large complex in wild type and *sprB* mutant cells, whereas it appeared to be part of a smaller complex in cells of *gldK, gldL, gldM,* and *gldNO* mutants (Fig. 5c). These results suggest that GldK, GldL, GldM, and GldNO contribute to multimerization and/or complex formation of GldJ, and may explain the absence of multirail structures in cells lacking these proteins.

**Presence of a multirail structure in the marine-gliding bacterium *S. grandis***. Aizawa[34,35] reported that bundle fibers are found in osmotically shocked cells of the marine-gliding Bacteroidetes *S. grandis*. *S. grandis* glides on solid surfaces at 5 μm per second[35,36]. *S. grandis*, a member of the class Sphingobacteriia[37], is not closely related to *F. johnsoniae*, a member of the class Flavobacteriia. Orthologs of *F. johnsoniae* Gld and Spr proteins were identified by BLAST search with the *S. grandis* genome[37] (Supplementary Table 3). *S. grandis* had orthologs of gliding motility proteins (GldA, GldB, GldD, GldF, GldG, GldH, GldJ, GldK, GldL, GldM, GldN, SprA, SprB, SprC, SprD, SprE, and SprT). However, orthologs of the periplasmic lipoprotein, GldI and the cell-surface lectin, RemA[38] (which is not essential for gliding) were not found in *S. grandis*. TEM analysis revealed the presence of abundant bundle fibers in osmotically shocked cells as previously reported[34,35] (Fig. 3k, l). Negative staining with PTA revealed thin filaments that may be SprB ortholog filaments, attached to the bundle fibers (Fig. 3l). The thickness of each fiber of the bundle fibers was 7.5 ± 0.8 nm ($N = 73$). Cryo-electron tomography also revealed the bundle fibers in intact *S. grandis* cells (Supplementary Movie 18). Structural similarity of the multirail structures (bundle fibers) between *F. johnsoniae* and *S. grandis* suggest that the multirail structure may be a common component of the gliding machinery in diverse members of the phylum Bacteroidetes.

**Discussion**

In a previous study, we proposed a model for gliding motility of *F. johnsoniae* where SprB is propelled along a closed helical-loop track on the cell surface[19]. Many of these SprB molecules are not engaged with the substratum and thus they move along the helical track without causing cell movement. In contrast, SprB molecules that attach to the substratum and are acted on by the motor result in rotation and translocation of the cell. Rotation of the gliding cells observed in this study support this model. CCW rotation of the gliding cells observed by the OMP domain-containing protein Fjoh_0697 (viewed from behind the rear of forward translocating cells) is consistent with the cell rotation generated by the left-handed helical movement of SprB. Shrivastava et al.[39] reported right-handed movements of SprB and cell movements. We do not have an explanation for this difference. Regardless of the handedness of the movements, it seems clear that SprB movement along helical tracks results in rotation and forward movement of gliding cells.

A detailed analysis of the SprB movement in this study shows that SprB can turn around in the middle, SprB can catch up with and overtake another SprB, and SprB can change its speed, suggesting that SprB does not always move at a constant speed on a single track. We observed the multirail structure associated with the SprB filaments in the *F. johnsoniae* cells that were burst by osmotic shock to reduce cell thickness and were then negatively stained with 0.5% uranyl acetate. It must be noted that we did not observe such multirail structures by cryo-electron tomography. Instead, parallel filamentous structures were observed in the periplasm of *F. johnsoniae* cells. The relationship between the

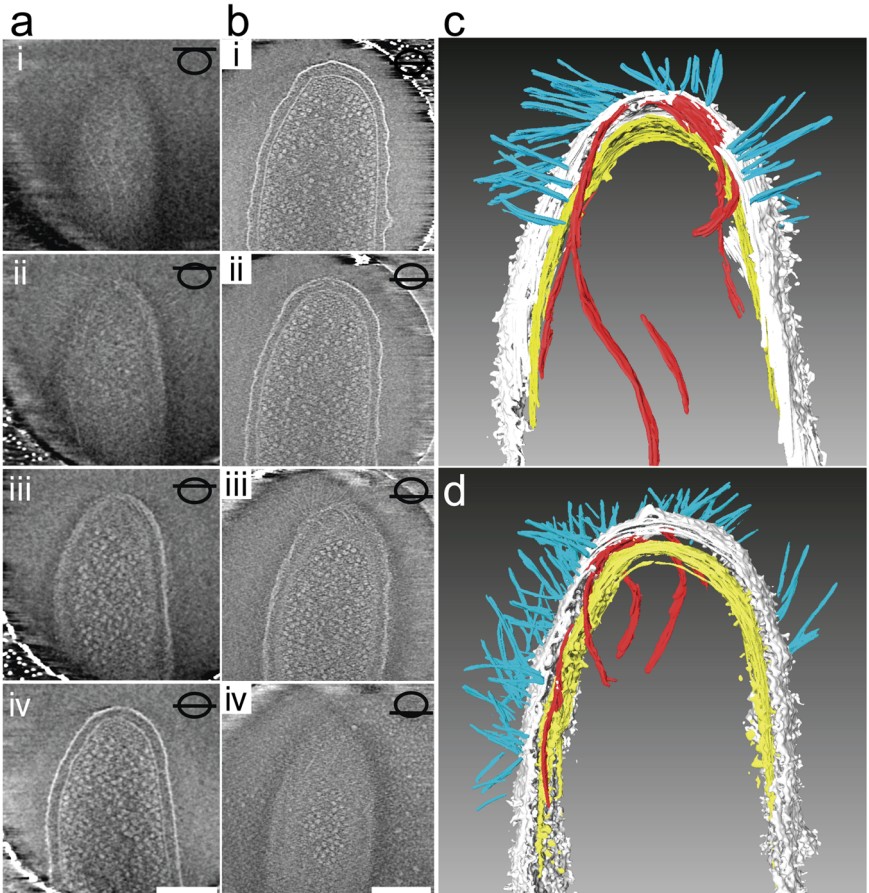

**Fig. 4 Cryo-electron tomography of wild-type *F. johnsoniae* cells.** Section images were taken from two representatives of 3D reconstructions of plunge-frozen wild-type cells (**a**, **b**). A schematic drawing with a slice position (inset) corresponds to each Z-slice image (i to iv in **a** and **b**). Bars indicate 200 nm. 3D segmentation images of tomograms shown in (**a**, **b**) (**c**, **d**, respectively) denote long thin electron-dense structures (red), SprB filament (cyan), peptidoglycan layer (yellow), and outer membrane (light gray).

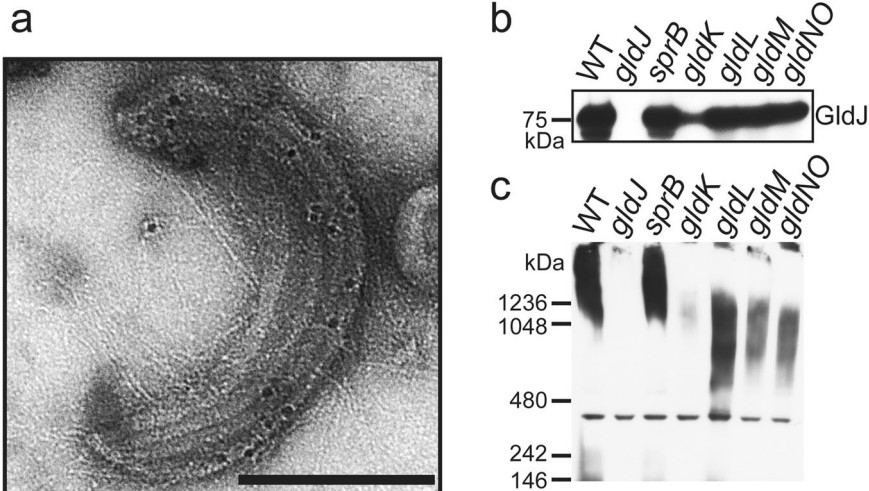

**Fig. 5 Association of GldJ with the multirail structure. a** Immunoelectron microscopy. Multirail structures peeled off of cells by osmotic shock were reacted with anti-GldJ antiserum followed by a gold-conjugated anti-rabbit antibody. Immunolabeled samples were stained with 0.5% uranyl acetate. Bars indicate 100 nm. **b**, **c** Immunoblots with anti-GldJ antiserum. Soluble extracts of *F. johnsoniae* wild type (WT), *gldJ*, *sprB*, *gldK*, *gldL*, *gldM*, and *gldNO* mutants were subjected to immunoblot analyses with anti-GldJ antiserum using SDS-PAGE (**b**) and Blue native-PAGE (**c**). Identical protein amounts were loaded in each lane. Protein band with a molecular mass of about 400 kDa in panel **c** was the result of nonspecific labeling by the antiserum since it was also present in the *gldJ* mutant lane, and thus serves as a loading control. Three technical replicates.

multirail structures and the filamentous structures is unknown. The location of the multirail structure on the periplasmic side of the outer membrane of *F. johnsoniae* was revealed by quick-freeze deep-etch electron microscopy. Considering this location, lipoproteins could be components of the structure. Immunoelectron microscopic analysis revealed that GldJ lipoprotein was associated with the multirail structure. It was previously reported that GldJ is organized in discrete bands that appear to form a helical structure[33], which is consistent with the present results. GldJ lipoprotein shares 30% identity with GldK lipoprotein over 382 amino acids[33]. *P. gingivalis* PorK (GldK homolog) with PorN makes a ring structure of ~8 nm in thickness and 52 nm in diameter[14,40]. The thickness of fibers in the multirail structure involving GldJ is almost the same as that of the PorK/PorN ring. Various *gld* mutants in addition to the *gldJ* mutant lacked the multirail structures. GldJ was reported to be unstable in *gldA*, *gldB*, *gldD*, *gldF*, *gldG*, *gldH*, and *gldI* mutants[33]. The lack of GldJ in these mutants may explain the absence of the multirail structures. GldJ appeared to be part of a large complex in wild-type cells, whereas it was present in lower molecular weight form in cells of *gldK*, *gldL*, *gldM*, or *gldNO* mutants. GldK, GldL, GldM, and GldN may be required for the assembly or stability of a GldJ complex associated with the multirail structures.

The marine-gliding bacterium *S. grandis* had a multirail structure similar to that of *F. johnsoniae* (Fig. 3e, l). The structure in *S. grandis* seemed to be continuous fibers and covered the entire cell body (Fig. 3k and Supplementary Movie 18). The presence of multirail structures in both *F. johnsoniae* and *S. grandis* cells provides insights into the mechanism of gliding motility employed by members of the phylum Bacteroidetes. We propose a helical multirail model for gliding motility in which the SprB filaments span the outer membrane, interacting with the substratum at the cell surface, and interacting directly or indirectly with the multirail structure in the periplasm (Fig. 6). The SprB filaments are propelled along the left-handed helical multirail structure. Sometimes the SprB filaments change direction even in the middle of a pole-to-pole path by shifting to another rail (Fig. 6). When motors act on SprB filaments that are firmly attached to the substratum, the filaments are propelled along the rails but remain stationary to the substratum. This results in rotation and forward movement of the cell body.

Gliding motility of the δ proteobacterium *Myxococcus xanthus* has been extensively studied, and it also involves cell rotation during migration[41–47]. The rotation of *M. xanthus* cells is CCW (viewed from the rear) as also observed here for *F. johnsoniae*. In *M. xanthus*, cytoplasmic and cytoplasmic membrane proteins migrate along a helical track in the cytoplasm, resulting in cell movement[44,47]. The cytoplasmic membrane motor proteins form a complex with periplasmic and outer membrane proteins that exert force on cell-surface adhesins that are attached to the substratum[46,48–50], which is similar to the *F. johnsoniae* motility model.

Although the helical movement of motility proteins drives the helical movement of both *F. johnsoniae* and *M. xanthus* cells[51]. There are numerous differences between the two systems. Many proteins have been identified that are involved in the gliding motility of both organisms but there is little if any similarity between them[2,3,52]. For *F. johnsoniae*, the cell-surface adhesin SprB appears to be propelled along a helical track associated with the periplasmic face of the outer membrane that may be comprised of the lipoprotein GldJ. There is no evidence for helical movement of components in the cytoplasm or cytoplasmic membrane and the current model involves motors, stationary on the cell, that propel SprB along the cell surface[23,24]. In contrast, for *M. xanthus* movement of cytoplasmic membrane proteins along a helical cytoplasmic track is proposed[44], resulting in movement of surface proteins. Gaps in our understanding of the *F. johnsoniae* and *M. xanthus* motility machines remain. For *M. xanthus* the cytoplasmic membrane motor proteins (AglR, AglQ, AglS) are responsible for force generation. In contrast, for *F. johnsoniae* the cytoplasmic membrane proteins GldL and GldM forms a nanoscale electrochemical motor[20,53–55], which drives both gliding motility and protein secretion. Control of the motors in response to the environment is also distinct for each organism. The *M. xanthus* motors are controlled by the *frz* chemotaxis system[56], whereas *F. johnsoniae* and other members of the Bacteroidetes lack critical chemotaxis proteins[2] and must use another mechanism to control cell movement.

Shrivastava et al.[57] suggested that the gliding motor in *F. johnsoniae* is rotary because *F. johnsoniae* cells rotated when tethered to glass by anti-SprB. We do not yet know how a rotary motor results in the helical movement of SprB. One possibility is that the rotating motor pushes on a tread carrying SprB filaments and propels it along the helical GldJ track[4,24]. Alternatively, the SprB filaments could be anchored on the tracks, and the tracks could be propelled by the motor[24,58].

Various unanswered questions regarding the Bacteroidetes gliding machinery remain to be solved. The proton gradient across the cytoplasmic membrane is required for SprB movement and cell gliding[19]. However, it is not known how the motor(s) transmit force to the SprB filaments on the cell surface. The apparent linkage of SprB filaments with the outer membrane-associated periplasmic rails described here may explain part of this transduction. Further studies including the clarification of the force-transducing mechanism are needed for a more complete understanding of this common form of Bacteroidetes motility.

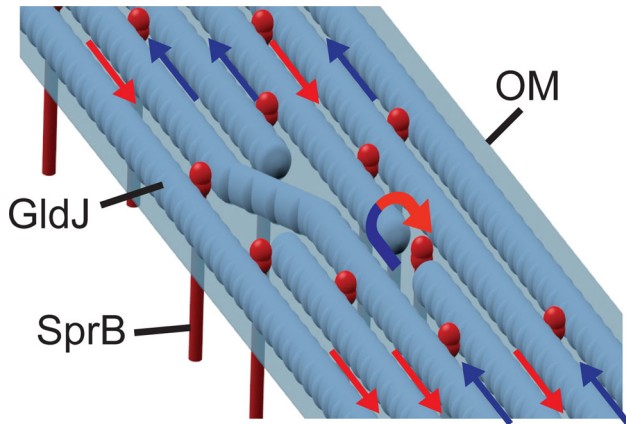

**Fig. 6 Models for the multirail structure for gliding machinery.** Multirail structure on the periplasmic face of the outer membrane (OM). The allows indicate the direction of SprB movement. The blue-to-red arrow indicates lane-switching by SprB resulting in a U-turn.

## Methods

**Bacterial strains and growth conditions**. Bacterial stains are listed in Supplementary Table 2. *F. johnsoniae* cells were grown in Casitone-yeast extract (CYE) medium at 25 °C with shaking[59]. To observe gliding motility, *F. johnsoniae* cells were grown in a motility medium (MM; 3-fold diluted CYE) to an optical density of around 1.0 at 600 nm. *S. grandis* cells were grown in Marine medium (Marine broth 2216 (BD, NJ, USA) with 0.5% tryptone (BD)) at 25 °C with shaking to an optical density of around 1.5 at 600 nm.

**Immunolabeling of cell-surface proteins**. For immunofluorescence microscopy of live cells, cells were harvested by centrifugation at $5000 \times g$ for 1 min, and the pellet was suspended in fresh MM with 1:500 to 1:1000 dilution of antiserum against SprB, against pooled cell-surface proteins or against Fjoh_0697 and incubated for 5 min. Cells were collected by centrifugation at $5000 \times g$ for 1 min and washed with fresh MM. The cells were suspended in fresh MM with 1:1000 dilution

of Alexa Fluor 555-conjugated antibody against rabbit IgG (Abcam, Cambridge, UK). After incubation for 5 min, cells were washed and suspended in fresh MM.

**Optical microscopy.** *F. johnsoniae* cells were poured into a tunnel slide assembled by using double-sided tape to attach a coverslip onto a glass slide[19,60], and were observed using an inverted microscope (Olympus IX83; Olympus, Tokyo, Japan). TIRF images were acquired with UAPoN 100×OTIRF objective lens (Olympus) using a 561 nm laser. Images were recorded with an iXon3 897 EMCCD camera (Andor Technology PLC, Northern Ireland, UK) using MetaVue software (Molecular Device, CA, USA). Time-lapse montage images and kymographs were processed with ImageJ 1.48r software (http://imagej.nih.gov/ij/).

**Isolation and identification of cell-surface proteins.** *F. johnsoniae* was grown in MM at 25 °C for 6 h. The cells were suspended in PBS (Sigma-Aldrich, MO, USA) and passed several times through a 26 G-1/2 inch needle to shear cell-surface structures. After the removal of the cells by centrifugation at $9000 \times g$ for 10 min, a fraction of surface proteins was collected by ultracentrifugation at $66,000 \times g$ for 60 min. Proteins were separated by SDS-PAGE and stained with Coomassie Brilliant blue R250. Protein bands were excised and digested by trypsin. Proteins were identified using matrix-assisted laser desorption ionization-time of flight mass spectrometry (MALDI-TOF MS; Urtraflex III, Bruker Daltonics, MA, USA).

**Osmotically shocked cells.** *F. johnsoniae* was grown in 5 ml of MM at 25 °C for 3 h. Cells were suspended in 100 μl of ice-cold sucrose solution (0.5 M sucrose, 0.15 M Tris-HCl, pH 7.5). The cells were placed on ice for 15 min and then mixed rapidly with 1.4 ml of ice-cold ultrapure water. Intact cells were removed by low-speed centrifugation at $9000 \times g$ for 3 min. Osmotically shocked cells were collected from the supernatant by high-speed centrifugation at $20,000 \times g$ for 5 min. For *S. grandis*, cells were first fixed with 1.5% paraformaldehyde for 30 min at room temperature and then washed twice with ice-cold ultrapure water or 50 mM $MgCl_2$ solution. Intact *S. grandis* cells were removed by centrifugation at $2000 \times g$ for 3 min and osmotically shocked cells were collected by centrifugation at $20,000 \times g$ for 5 min. Centrifugation was done at 4 °C.

**Electron microscopy.** Samples were negatively stained with 2% PTA (pH 7.0) or 0.5% uranyl acetate on a Butvar B-98 (Sigma-Aldrich) coated copper grid, and observed by transmission electron microscopy (JEM-1230NT; JEOL, Tokyo, Japan). Micrographs were taken at an accelerating voltage of 80 kV. For immunogold electron microscopy, osmotically shocked cells were treated with 1000-fold diluted antiserum against SprB protein in PBS containing 2% BSA and incubated on ice for 20 min. The cells were washed three times with PBS and treated on ice with 20-fold diluted goat anti-rabbit IgG conjugated to 5 nm diameter gold particles (BBI solutions, Cardiff, UK) in PBS containing 2% BSA for 20 min, washed three times, and then stained with 0.5% uranyl acetate. For anti-GldJ immunostaining, osmotically shocked cells were fixed with 1.5% paraformaldehyde before treatment with anti-GldJ rabbit polyclonal antiserum[33].

**Preparation of quick-freeze deep-etch replica specimens.** Bacterial cells, which were partly disrupted by osmotic shock, were mounted onto a cover glass and quickly frozen by metal contact at liquid-nitrogen temperature. Frozen samples were subjected to freeze-fracture deep-etch replication[61,62]. Samples were knife-fractured, deep-etched at −104 °C for 10 min, rotary-shadowed with Pt/C at an angle of 20°, and then backed with pure carbon by a freeze-fracture device (JFD-V: JOEL). Replicas were floated off the cover glass onto the surface of full-strength hydrofluoric acid. Household bleach was occasionally used to remove the remaining debris from the replica. Replicas were rinsed with three changes of water and picked up onto copper grids for electron microscopic examination.

**Cryo-electron tomography.** Quantifoil molybdenum 200 mesh R0.6/1.0 grids (Quantifoil Micro Tools GmbH, Großlöbichau, Germany) were glow discharged and pretreated with a solution of 10 nm colloidal gold particles concentrated 1.5 times before use (MP Biomedicals, CA, USA) for tomogram alignment. A 3 μl sample was applied to the grid, blotted with filter paper, and plunged into liquid ethane using Vitrobot (FEI, OR, USA). Images were collected at the liquid-nitrogen temperature using a Titan Krios FEG transmission electron microscope (FEI) operated at 300 kV on FEI Falcon 4 k × 4 k direct electron detector (FEI). The magnification was calibrated by measuring the layer-line spacing of 23.0 Å in the Fourier transform of images of tobacco mosaic virus mixed in the sample solution. The pixel size on the specimen was 0.57 nm. Single-axis tilt series were collected covering an angular range from −70 to 70 with a nonlinear Saxton tilt scheme at 4–10 μm underfocus using the Xplore 3D software package (FEI). A cumulative dose of 200 e−/Å² or less was used for each tilt series. Images were generally binned two-fold and 3D reconstructions were calculated using the IMOD software package[63]. Surface-rendering images were obtained using the three-dimensional modeling software Amira 5.2.2 (Visage Imaging, San Diego, CA).

**Blue native gel electrophoresis.** Cells were sonicated in lysis buffer containing 1% (weight per volume) *n*-dodecyl-β-D-maltoside (DDM), 0.5 M sucrose, 10 mM Tris-HCl (pH 7.5), 1 mg per ml DNase, and protease inhibitor cocktail (Sigma-Aldrich). Soluble fractions were mixed with 10× Blue native sample buffer (5% CBB G-250, 500 mM 6-aminocaproic acid, 100 mM Bis-Tris-HCl, pH 7.0), and loaded on a native gel (3–12% Bis-Tris gel; Thermofisher scientific, MA, USA). Proteins on the gel were blotted on PVDF membrane and subjected to immuno-detection with anti-GldJ rabbit polyclonal antiserum.

**Construction of *F. johnsoniae* ΔsprD and ΔsprF mutants.** A 1.9-kb fragment downstream of and spanning the final 33 bp of *sprD* was amplified using primers 1296 (5'-GCTAGGTCGACGGGAAAATGGCAATCGTAAAAG-3', SalI site underlined) and 1297 (5'-GCTAGGCATGCAGGAGTTGGCGACGAATCTC-TAATG-3', SphI site underlined). The fragment was digested with SalI and SphI and ligated into pRR51[64], which had been digested with the same enzymes, to generate pYT55. A 2.1-kb fragment spanning *sprC* and the first 57 bp of *sprD* was amplified using primers 1294 (5'-GCTAGGGATCCTCAACCCTAAAAAGCCA-GACTACAG-3', BamHI site underlined) and 1295 (5'-GCTAGGTCGACATA-GAGTAAACATGAAAAACCGCAG-3', SalI site underlined). The fragment was digested with BamHI and SalI and fused to the downstream region of *sprD* by ligation with pYT55, which had been digested with the same enzymes, to generate the deletion construct pYT58. pYT58 was introduced into *F. johnsoniae* CJ1827 by triparental conjugation to construct the ΔsprD mutant CJ2246. To delete *sprF*, pYT313[65] containing 1.8-kb regions upstream and downstream of *sprF* was introduced into *F. johnsoniae* CJ1827 by triparental conjugation to construct the ΔsprF mutant CJ2518.

**Statistics and reproducibility.** Micrographs and electron micrographs in Figs. 1, 2, 3, 5 and Supplementary Figs. 1–3 show representative examples from experiments that were repeated independently at least three times with similar results. Tomograms in Fig. 4 show representative examples from experiments that were repeated independently at least twice with similar results. Immunoblots and protein electrophoresis in Fig. 5 and Supplementary Fig. 1 were repeated independently at least three times with similar results. Sample sizes for the results represented box plots and dot plots in Figs. 1 and 2 are indicated in each figure panel. Box plots in Figs. 1d and 2h represent the minimum value, 25th percentile, median, 75th percentile, and maximum value.

**Reporting summary.** Further information on research design is available in the Nature Portfolio Reporting Summary linked to this article.

## Data availability

All data generated during this study are included in this article and its supplementary materials. Uncropped images of the immunoblots and gel in the figures (Fig. 5b, c and Supplementary Fig. 1a, b) are shown in Supplementary Fig. 4. Source data underlying Figs. 1d–f and 2h are presented in Supplementary Data 1 and 2, respectively.

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

## Acknowledgements

We thank S. Aizawa for supplying *S. grandis* strain, T. Hamaguchi for helping to draw the gliding model in Fig. 6, and the general supporting team at Osaka City University for Scientific Research on Innovative Areas Harmonized Supramolecular Motility Machinery and Its Diversity supported by the Japan Society for the Promotion of Science (JSPS) Kakenhi Grant (Grant ID 25117501), directed by M. Miyata, for technical help with electron microscopy. This work was supported by the JSPS Kakenhi Grants (Grant IDs

24117006 and 25293375 to K.N. and 17K17085 to S.S.) and by National Science Foundation Grant MCB-1516990 to M.J.M.

## Author contributions

S.S. and K. Nakayama designed research; S.S., Y.O.T., E.K., A.K., T.K., D.N., and Y.Z. performed research; S.S., E.K., T.K., M.M., D.N., K. Namba, M.J.M., and K. Nakayama analyzed the data; and S.S., D.N., M.J.M., and K. Nakayama wrote the paper.

## Competing interests

The authors declare no competing interests.
