## [Peer Review File · Communications Biology]

This manuscript has been previously reviewed at another Nature Portfolio journal. This document only contains reviewer comments and rebuttal letters for versions considered at Communications Biology.

Reviewers' comments:

Reviewer #1 (Remarks to the Author):

To be clear, the multi-rail structures were observed in the bubblelike structure on the side of the cell or in the membrane vesicles. Although the authors claimed the multi-rail structures in the WT cell(Fig. 3AB&H), it is not as convincing as others. In addition, with my experiences and knowledge of cryoet, for such striking features of multi-rail structures, it should be visualized by cryoet along the cell body if the authors imaged 10 to 20 cells. Instead, by cryoet, the authors observed the "parallel filamentous structures", in fact, from the images(Fig. 4), there are two parallel filaments with a spacing around 50 nm. As the comments from reviewer #2, it is hard to believe the parallel filamentous structures are the same as multi-rail structures. In the revised MS, the authors retracted this statement, but it is still misleading, I suggest the authors should clearly state the multi-rail structures were visualized from the bubbles of the cell or the membrane vesicles, especially in the texts where cited Fig. 3C-G. I suggest the authors consider the reviewer #2's comments seriously.

Other comments:

In Fig. 3, each image is in different scales, I suggest reducing to two scales, making it easy to compare.

Reviewer #1' Comments

To be clear, the multi-rail structures were observed in the bubblelike structure on the side of the cell or in the membrane vesicles. Although the authors claimed the multi-rail structures in the WT cell(Fig. 3AB&H), it is not as convincing as others. In addition, with my experiences and knowledge of cryoet, for such striking features of multi-rail structures, it should be visualized by cryoet along the cell body if the authors imaged 10 to 20 cells. Instead, by cryoet, the authors observed the “parallel filamentous structures”, in fact, from the images(Fig. 4), there are two parallel filaments with a spacing around 50 nm. As the comments from reviewer #2, it is hard to believe the parallel filamentous structures are the same as multi-rail structures. In the revised MS, the authors retracted this statement, but it is still misleading, I suggest the authors should clearly state the multi-rail structures were visualized from the bubbles of the cell or the membrane vesicles, especially in the texts where cited Fig. 3C-G. I suggest the authors consider the reviewer #2's comments seriously.

Response to the reviewer #1's comments

Following the reviewer's suggestion, we modified the manuscript as follows.

In the Title: *Filamentous Structures in the Cell Envelope Associated with Bacteroidetes Gliding Machinery.*

In the Results part (L 235-237): *However, multi-rail structures similar to those observed in osmotically burst cells by TEM and quick-freeze deep-etch electron microscopy were not observed by cryo-electron tomography.*

In the Discussion part (L 293-296): *It must be noted that we did not observe such multi-rail structures by cryo-electron tomography. Instead, parallel filamentous structures were observed in the periplasm of F. johnsoniae cells. The relationship between the multi-rail structures and the filamentous structures is unknown.*

In addition, in the descriptions of other places (L 40, 43, 75, 159, 182, 201, 212, 221, and 226), we only pointed out the possibility of the existence of multi-rail structures.

Other comments:

In Fig. 3, each image is in different scales, I suggest reducing to two scales, making it easy to compare.

Response to the comment

Following the reviewer's suggestion, Panels *a*, *c*, and *k*, and Panels *b*, *d*, *e*, *f*, *g*, *h*, *i*, and *l* in Fig. 3 except Panel *j* are each scaled to the same scale.

REVIEWERS' COMMENTS:

Reviewer #1 (Remarks to the Author):

I suggest to accept the revised MS.